# Dynamically Temperature-Voltage Controlled Multifunctional Device Based on VO_2_ and Graphene Hybrid Metamaterials: Perfect Absorber and Highly Efficient Polarization Converter

**DOI:** 10.3390/nano9081101

**Published:** 2019-08-01

**Authors:** Min Mao, Yaoyao Liang, Ruisheng Liang, Lin Zhao, Ning Xu, Jianping Guo, Faqiang Wang, Hongyun Meng, Hongzhan Liu, Zhongchao Wei

**Affiliations:** Guangdong Provincial Key Laboratory of Nanophotonic Functional Materials and Devices, School of Information and Optoelectronic Science and Engineering, South China Normal University, Guangzhou 510006, China

**Keywords:** vanadium dioxide, phase change material, graphene, perfect absorber, polarization converter

## Abstract

Vanadium dioxide (VO_2_) is a temperature phase change material that has metallic properties at high temperatures and insulation properties at room temperature. In this article, a novel device has been designed based on the dielectric metasurface consisting of VO_2_ and graphene array, which can achieve multiple functions by adjusting temperature and voltage. When the temperature is high (340 K), the device is in the absorption state and its absorptivity can be dynamically controlled by changing the temperature. On the other hand, the device is in the polarization state under room temperature, and the polarization of electromagnetic waves can be dynamically controlled by adjusting the voltage of graphene. This device can achieve a broadband absorber (the maximum absorptance reaches 99.415% at wavelengths ranging from 44 THz to 52 THz) and high polarization conversion efficiency (>99.89%) in the mid-infrared range, which has great advantages over other single-function devices. Our results demonstrate that this multifunctional device may have widespread applications in emitters, sensors, spatial light modulators, IR camouflages, and can be used in thermophotovoltaics and wireless communication.

## 1. Introduction

In recent years, with the development of phase change materials, it has been found that the optical properties of VO_2_ phase change material change with temperature. Therefore, VO_2_ materials are gradually applied to achieve dynamic tunability of optical devices [1,2,3] and RF electronics [4,5]. The VO_2_ phase transition is generally considered as an insulator-metal phase transition, which includes the insulator state, metallic state and the transition state between them. When the temperature of VO_2_ changes from lower to higher than the phase transition temperature, the lattice of VO_2_ will be twisted from monoclinic phase structure of insulator state to rutile tetragonal phase structure of metallic state, which is reversible and accompanied by the change of energy band structure of VO_2_. Although the phase transition process of VO_2_ is very short, there is a gradual process in which the electromagnetic properties of VO_2_ will change significantly [6].

Graphene is an optical material with tunability. With the relativistic-like linear energy dispersion in graphene, electrons can travel at a Fermi velocity merely 100 times slower than light speed, and those are the unique electronic and optical properties of graphene. The conductivity of graphene in universal optics is *e*^2^/4*ℏ* where *ℏ* is reduced Planck constant and *e* is electronic charge. The graphene is a semimetal that does not have as much free charge as metal. In general, the free charge concentration of graphene can be changed through chemical doping or bias voltage. Therefore, the semimetallic characteristics of graphene, allows for electrical tunability that conventional metals cannot realize.

In this article, a hybrid metamaterial based on VO_2_ and graphene is designed to achieve perfect absorption and polarization conversion function simultaneously. Among micro-nano optoelectronic devices, broadband perfect absorber and high efficiency polarization converters have widespread applications, such as thermal emitters, imaging devices, sensors, modulators and camouflage devices, etc. [7,8,9,10,11] They have become the research hotspot in the mid-infrared band [9,12,13,14,15]. However, traditional perfect absorbers and polarization converters mostly use complex structures to achieve the corresponding functions. Most perfect absorbers use structural characteristics to achieve perfect absorption, while polarization converters achieve polarization by digging holes of different shapes in metal [16,17,18,19]. For example, the tunable ultra-broadband terahertz absorber has been reported by Xu et al. [20], which uses multiple layers of graphene Ribbons. Besides, Guo et al. realized the ultra-broadband infrared metasurface absorber by adopting multi metallic layers, but their work cannot accomplish tunability [21]. These proposed structures can only be tuned by changing geometric parameters of the structures. The applications of the structures are limited and active control of the spectral response is needed. What’s more, conventional micro-nano photonic devices only achieve single function, limiting their utilization, but devices designed in this paper consist of VO_2_ and graphene achieve different functions at different temperatures. At the high temperature (340 K), VO_2_ exhibits metallic properties. With reasonable design of the VO_2_ and selection of material parameters, the electromagnetic component of the incident electromagnetic waves can be coupled, so that the electromagnetic wave incident on the surface of the metamaterial in a specific frequency band can produce neither reflection nor transmission, thus near-unity 100% perfect absorption is obtained. The advantage of the absorber is that it can work in a high temperature environment, and as the temperature increases, the absorption effect of the absorber to the electromagnetic wave will be enhanced instead of being weakened. In low temperature conditions (e.g., room temperature 300 K), VO_2_ is at an insulator state. Under this condition, the polarization of the electromagnetic wave is realized and the proposed device is capable of rotating a linear polarization state into its orthogonal one by tuning the Fermi level of graphene in the hybrid metamaterial.

## 2. Material Model Analysis

When the temperature is switched from lower to higher than 68 °C, the VO_2_ experiences a phase transition from insulator to metallic state. This phase change causes a dramatic alteration of the optical properties of VO_2_ in mid-infrared wavelengths. For VO_2_ insulator material, assume that its permittivity is *ε_d_*, there are random metallic particles in it, and its permittivity is *ε_m_*. When the volume fraction V (more than 20%) of the metallic particles is relatively large, the space between the metallic particles is relatively small, so the interaction between particles cannot be ignored. Then the permittivity of VO_2_ can be expressed by the simple Bruggeman theory [22,23,24,25].
(1)$$\varepsilon_{{(\mathrm{VO}}_{2})}\;=\;\frac{1}{4}\left\{\varepsilon_{d}{(2\;-\;3V)\;+\;\varepsilon }_{m}(3V\;-\;1)\;+\;\sqrt{\varepsilon_{d}{(2\;-\;3V)\;+\;\varepsilon }_{m}{\left(3V\;-\;1\right)}^{2}{\;+\;8\varepsilon }_{d}\varepsilon_{m}}\right\}$$

The permittivity of the metallic components in VO_2_ materials can be expressed by the Drude model [2,6]
(2)$$\varepsilon (\omega )\;=\;\varepsilon_{\infty }\;-\;\frac{\omega_{p}^{2}(\sigma )}{\omega (\omega \;+\;i\gamma )}$$
where $\varepsilon_{\infty }\;=\;12$ is the permittivity at high frequency, $\omega_{p}(\sigma )$ is the conductivity dependent plasmon frequency and *γ* is the collision frequency. Besides, both $\omega_{p}(\sigma )$ and *σ* are proportional to free carrier density. Therefore, the frequency of plasmon at *σ* can be roughly expressed as $\omega_{p}^{2}(\sigma )\;=\;\frac{\sigma }{\sigma_{0}}\omega_{p}^{2}(\sigma_{0})$. According to reference, $\sigma_{0}\;=\;3\times 10^{5}\;S/m$, $\omega_{p}^{}(\sigma_{0})\;=\;1.4\times 10^{15}\;\mathrm{rad}/s$, $\gamma \;=\;5.75\times 10^{13}\;\mathrm{rad}/s$. Also, we demonstrate a polarization based on graphene. The permittivity of gold following a Drude model [26], $\varepsilon_{\mathrm{Au}}\;=\;\varepsilon_{\infty }\;-\;\omega_{p}/(\omega^{2}\;+\;i\omega \Gamma )$ with plasma frequency $\omega_{P}=1.3\times 10^{16}\;\mathrm{rad}/s$, damping constant $\Gamma =1.11\times 10^{14}\;\mathrm{rad}/s$ and $\varepsilon_{\infty }\;=\;1.53$ [27].

The surface dynamic conductivity of graphene can be approximately written as follows [28]:(3)$$\begin{array}{ll} \sigma (\omega ) & =\sigma_{intra}(\omega )\;+\;\sigma_{inter}(\omega )\\ & =\frac{2e^{2}k_{B}T}{\pi \hbar (\omega \;+\;i\tau^{-1})}\ln\left[2\cosh(\frac{E_{F}}{2k_{B}T})\right]\;+\;\frac{e^{2}}{4\hbar }\left[\frac{1}{2}\;+\;\frac{1}{\pi }\arctan(\frac{\hbar \omega \;-\;2E_{F}}{2k_{B}T})\;-\;\frac{i}{2\pi }\ln\frac{{\left(\hbar \omega \;+\;2E_{F}\right)}^{2}}{{\left(\hbar \omega \;-\;2E_{F}\right)}^{2}\;+\;4{\left(k_{B}T\right)}^{2}}\right] \end{array}$$
where *e* is the charge of an electron, *k_B_* is the Boltzmann constant, *T* is the temperature, *τ* is relaxation rate, *ω* is the frequency of incident light and *E_F_* is the Fermi level of graphene.

The first term in Equation (3) is derived from the intra-band transition and the second term is derived from the inter-band transition. In the terahertz to mid-infrared range, the probability of electron-induced inter-band transitions in the graphene is low due to the limited energy of the incident photons. Here we only consider the case of highly doped graphene, which satisfies the conditions of $E_{F}\;\gg \;k_{B}T$ and $E_{F}\;\gg \;\hbar \omega$ at room temperature (300 K). In this case, the conductivity term of the intra-band transition can be approximated to the Drude model [29]:(4)$$\sigma (\omega )\;\approx \;\frac{e^{2}E_{F}}{\pi \hbar^{2}}\frac{i}{\omega \;+\;i/\tau }$$

## 3. Design of Structure

The structural model is shown in Figure 1, and the optimized parameters of each part of the structure are marked in Table 1. Figure 1a shows the 3D schematic of the multifunctional device. Here, the structural periodicity, thickness of the dielectric and thickness of the back reflector are fixed at *p*, *d* and *t*, respectively. The thickness of the top VO_2_ layer and the diameter are *h* and *D*. A rectangular hole of *l_x_* × *l_y_* size was dug on the surface of the graphene, and the rectangular hole was rotated 45° clockwise. The incident polarization is assumed to be along the *x*-axis in the simulation. In the simulations, the simulation region has a size of 1.57 µm × 1.57 µm, the mesh size in VO_2_ is Δ*x* = Δ*y* = 0.015 µm, and Δ*z* = 0.005 µm. The simulation time and mesh accuracy were set “20,000 (fs)” and “4”, respectively. The boundary condition along the *z* direction was the perfectly matched layer (PML), and those for *x* and *y* directions were the Periodic.

### 3.1. VO_2_-Based Tunable Metamaterial Absorber

The proposed metasurface multifunction device is composed of VO_2_, a graphene layer, a dielectric layer and the bottom metal reflector, as shown in Figure 1a. At a temperature of 340 K, the lattice of VO_2_ will be twisted from low-temperature monoclinic phase structure of insulator state to rutile tetragonal phase structure of metallic state [22]. At the same time, VO_2_ is transformed from insulator state to metallic state. Phase transitions cause change of conductivity of VO_2_ by several orders of magnitude and strong changes in optical properties. Also, in the mid-infrared range, the Pauli-blocking occurs in the doped graphene, and the optical conductivity is minimal, resulting the absorptance of graphene in the mid-infrared band being less than 2.3% [8]. Therefore, in the mid-infrared absorption state, the influence of doped graphene on the device can be ignored.

In order to prove the tunable absorption characteristics of VO_2_, we studied the absorption performance of the device by the finite-difference time-domain (FDTD) method (FDTD solutions V8.19, Lumerical Inc., Vancouver, BC, Canada). According to Kirchhoff’s rule, the sum of absorptance A, transmittance T and reflectance R should be equal to 1 (A + T + R = 1). Due to the fact that the thickness of the bottom Au material larger than its largest skin depth of δ = λ/[2πIm(nAu)] ≈ 28 nm [30] in the mid-infrared frequencies, the transmittance T≈0, the absorptance can be calculated as A = 1 − R. Figure 2 shows the absorptance when increasing the conductivity of VO_2_ from 10 Ω^−1^ cm^−1^ to 3000 Ω^−1^ cm^−1^.

It can be seen from Figure 2 that as the conductivity of VO_2_ increases, the absorptance spectra of the device change significantly. More specifically, the absorptance spectra increase with the increment of the conductivity of VO_2_. When the conductivity of VO_2_ is 3000 Ω^−1^ cm^−1^, the absorptance reaches 99.415% and the full width at half maximum (FWHM) of the absorption peak is 14.39 THz with a central frequency of 48.078 THz. With the existence of the interference of the fields between metallic VO_2_ and dielectric layer, the perfect absorptance is realized [31]. Figure 3 demonstrates that this phenomenon is primarily caused by the change of the permittivity of VO_2_. When the conductivity of VO_2_ changes from 10 Ω^−1^ cm^−1^ to 3000 Ω^−1^ cm^−1^, the real and imaginary parts of the permittivity increase rapidly in the range of 0 to 80 THz, resulting in VO_2_ undergoing an optical transition from insulator to metallic state. Therefore, it is concluded that the larger the conductivity of VO_2_, the stronger the metallicity of VO_2_, and the closer the absorber formed by VO_2_ to near-unity absorption.

We use the Fabry–Perot theory to explain the mechanism of VO_2_ metamaterial perfect absorber. The structure of this device can generally be regarded as a Fabry–Perot resonator which is consisting of a partially reflecting mirror and a fully reflecting mirror. Figure 4 illustrates the optical coupling in such a Fabry–Perot resonator. The electromagnetic wave is incident vertically along the *x*-axis polarization direction. Assuming that the incident electromagnetic wave amplitude is *E_inc_* and the reflected electromagnetic wave amplitude is *E_ref_*, then the air-metasurface interface can be written as follows [7]:(5)$$R\;=\;\frac{E_{ref}}{E_{inc}}\;=\;\frac{r_{12}\;+\;(t_{12}t_{21}\;-\;r_{12}r_{21})r_{23}e^{i2\beta d}}{1\;-\;r_{21}r_{23}e^{i2\beta d}}$$
where the *r*_12_, *r*_21_ (*t*_12_, *t*_21_) are the ratios of the complex electric field amplitude of reflected waves (transmitted wave) to that of incident waves at the interface of air (metal mirror). The reflection coefficients of the bottom metal mirror are *r*_23_ = −1, *β =* 2*πn*_2_/*λ*_0_ is the propagation constant and *n*_2_ is the refractive index of the dielectric. According to A = 1 − R, when R = 0, the absorptivity reaches the maximum, which satisfies the critical coupling condition [32].

To further study the absorption properties of the absorber, Figure 5 analyses the distribution of normalized electric field intensity (|*E*|) and magnetic field intensity (|*H*|) at *f* = 48.0781 THz. Figure 5a,b are the electric field intensity distributions on *xy* plane at the VO_2_ film interface, from the air and from the dielectric substrate, respectively. Figure 5c are the electric field intensity distributions on *xz* plane in the middle of the cell at 48.0781 THz. Figure 5d are the magnetic field intensity distributions on *xz* plane in the middle of the cell at 48.0781 THz. From the Figure 5a–c, a very strong electric field around curved edge of VO_2_ which corresponds to maximum absorption is found at 48.0781 THz. It exhibits that stronger electric field confinement will lead to higher absorption. In addition, most of the electric fields are confined to the curved edge of the VO_2_ film owing to the localized surface plasmon resonance (LSPR) of the metallic VO_2_.

Polarization independence and incident angle insensitivity are important factors which should be considered in practical field. To investigate the polarization independence and larger incident angle insensitivity of absorber, we study absorption performances with different polarization angles and incident angles. From the Figure 6a,b, when the variations of incident angles range from 0° to 40°, near-unity absorptions at resonance frequency are still to be achieved. As shown in the Figure 6c, the absorption spectra remain unchanged with the variations of polarization angles from 0° to 90° under normal incidence because of the highly symmetrical structure. Therefore, we can consider that this tunable absorber is irrelevant to polarization and insensitive to large incident angle, which will make the absorber widely used in practical field.

### 3.2. Graphene-Based Tunable Metamaterial Polarization Converter

At room temperature (300 K), VO_2_ is in an insulator state. The relative permittivity of VO_2_ is 9, while the conductivity in the insulator state is smaller than 200 S/m [33,34]. Under this condition, the polarization of the electromagnetic wave can be realized by tuning the Fermi level of the graphene, thereby changing the device from a perfect absorber to a polarization converter. The Fermi level of graphene is mainly achieved by adjusting the voltage, while the top graphene plays the role of gate electrode. When a bias voltage is applied to the top graphene, the carrier concentration in the graphene can be dynamically controlled, and then Fermi level of graphene can be under control. The Fermi level $E_{F}$ in graphene is related to the bias voltage $V_{g}$ as follows [35,36]
(6)$$E_{F}\;\approx \;\hbar v_{f}\sqrt{\frac{\pi \varepsilon_{r}\varepsilon_{0}V_{g}}{ed}}$$
where $v_{f}$ is the Fermi velocity (1.0 × 10^6^ m/s). $\varepsilon_{r}$ and $\varepsilon_{0}$ are the dielectric permittivity and vacuum permittivity, respectively.

The designed polarization converter can achieve high polarization conversion rate (PCR) in wide wavelength range. In the simulation, the plane wave polarized along the *x*-axis is perpendicularly incident on the graphene surface. The reflection coefficients of *x*- and *y*-polarized reflected waves are defined as co-polarization reflection coefficient *R_xx_* and cross-polarization reflection coefficient *R_xy_* (*R_ij_* denotes *j*-polarized reflection from *i*-polarized incidence), respectively. The polarization converter ratio is defined as $\mathrm{PCR}\;=\;R_{xx}^{2}/(R_{xx}^{2}\;+\;R_{xy}^{2})$. And the reflection phase difference between *R_xx_* and *R_xy_* is deemed as *φ_xy_* = arg(*R_xx_*) − arg(*R_xy_*) and *φ_xy_* can be any value within [−*π*,*π*].

Figure 7a,b show the relationship between PCR and phase difference as Fermi level and frequency. We saw an increase in the Fermi level of graphene from 0.15 eV to 0.95 eV. In Figure 7a, while the Fermi level is 0.95 eV in the frequency range of 6 to 22 THz, the polarization converter ratio reaches a maximum of 99.89%. Figure 8 shows the normalized electric field intensity and magnetic field intensity at the frequency of 13.4177 THz. It can be observed that the electric resonance is mainly concentrated on the corners of each rectangular hole, indicating a strong coupling between neighboring holes. We analyze the physical mechanism of proposed graphene cross polarization converters by electric field distribution, which is shown in the inset of Figure 8.

In this section, we still use the Fabry–Perot model to explain the broadband linear polarization modulation of electromagnetic waves. We decompose the electromagnetic waves incident in the *x* direction into two perpendicular components along the *u*, *v* direction, which correspond to the long and short axis of the resonator, respectively (As shown in Figure 9a). The simulated amplitude and phase of the reflection are shown in Figure 9b,c. Figure 9b,c show that the amplitude of *u* and *v* directions are substantially the same and the relative phase difference is *π*, which results in polarization rotation of 90 degrees. The structure can convert the incident linear polarized light into cross linear polarized light in a wide range. The high polarization conversion rate can be explained by the interference theory of F-P cavity. The electromagnetic wave vertically incident to the metasurface enters the F-P cavity, in which the multi-reflection superposition results in destructive and constructive interference of the co-polarized and cross-polarized light, respectively [37,38].

## 4. Experimental Feasibility

The experimental measurement is shown in Figure 10, the multifunctional device is placed on the temperature controller, in this case, the phase state of VO_2_ can be switched between metallic and insulator state through temperature controller, and the reflectance and absorptance of the device at different temperature can be measured via the optical detector. Therefore, we can change the temperature and measure the influence of conductivity change on the absorptance in metallic state, on the contrary, we can keep the temperature of the device at room temperature to make VO_2_ in an insulator state, in such a case, the device serves as a polarization converter, its polarization state can be adjusted by bias voltage of graphene. We can use the linear polarizer to select the component—*R_xx_* and *R_xy_* of the reflected light at different Fermi levels for the power measurement, respectively, thus finally their according PCR can be calculated.

## 5. Conclusions

We have demonstrated a novel device design based on the dielectric metasurface consisting of VO_2_ and graphene array, which can achieve multiple functions by adjusting temperature. In the absorption state, the simulated results demonstrate that the absorption coefficient of over 90% can be realized in the spectral range of 44 to 52 THz (the maximum absorptance reaches 99.415%). The absorption spectrum can be adjusted actively by dynamically altering the conductivity of VO_2_. In addition, this type of metamaterial perfect absorber is relatively insensitive to polarization angle. In the state of polarization, the polarization converter can achieve broadband polarization in mid-infrared band, and its maximum polarization conversion rate can reach 99.89%. The operating bandwidth and magnitude of the PCR can be tuned easily by adjusting the Fermi level of graphene. In summary, the wideband, high FWHM and high PCR demonstrate that the proposed multi-function device can be applied in many promising fields, such as smart absorbers, photovoltaic devices and tunable polarization converter. In a way, we believe that multifunctional devices consisting of VO_2_ and graphene can greatly save manufacturing costs, as well as make the device manufacturing easier, more convenient and faster.

## Figures and Tables

**Figure 1 nanomaterials-09-01101-f001:**
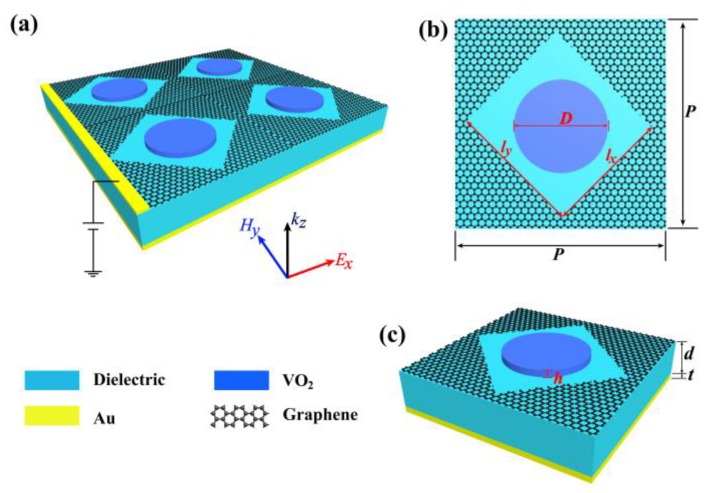
(**a**) Schematic diagram of multifunctional device structure. (**b**) Structural top view. (**c**) Unit cell of the structure.

**Figure 2 nanomaterials-09-01101-f002:**
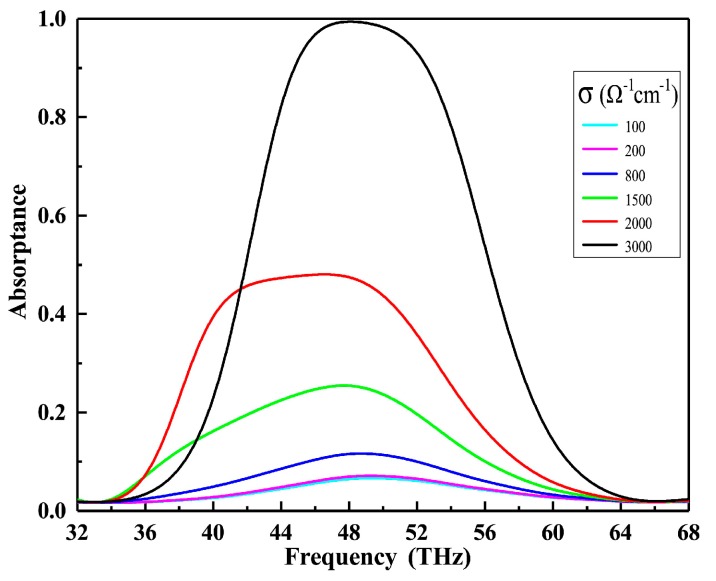
Absorptance spectra of multi-functional device under different conductivity of VO_2_ under normal incidence.

**Figure 3 nanomaterials-09-01101-f003:**
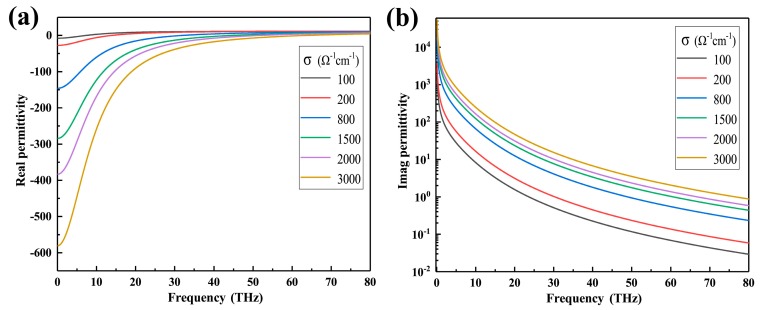
The calculated permittivity of VO_2_ at a series of conductivities. (**a**) Real part of permittivity of VO_2_. (**b**) Imaginary part of permittivity of VO_2_ (logarithmic coordinates for Y axis).

**Figure 4 nanomaterials-09-01101-f004:**
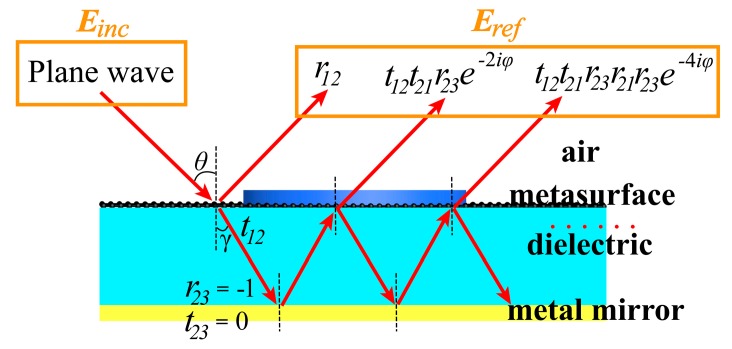
Schematic of the multi-reflection under a normally incident x-polarized wave.

**Figure 5 nanomaterials-09-01101-f005:**
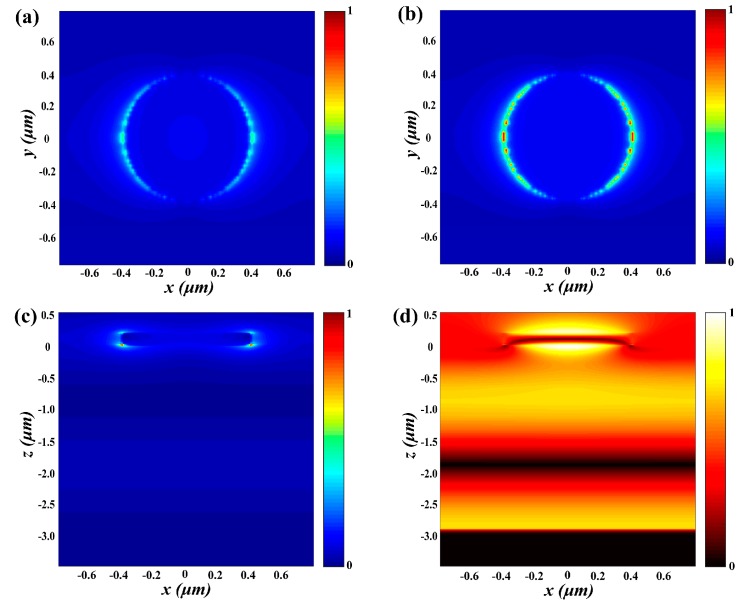
(**a**) |*E*| of the absorber at the interface between air and VO_2_ film (**b**) |*E*| of the absorber at the interface between VO_2_ and dielectric layer (**c**) |*E*| in the center of the *xz* plane of the absorber of the cell (**d**) |*H*| at the *xz* plane of the absorber. All frequencies are set at 48.0781 THz.

**Figure 6 nanomaterials-09-01101-f006:**

Tuned the incident angles from 0° to 40° with (**a**) TM mode (**b**) TE mode. (**c**) Variations of polarization angle within from 0° to 90° with normal incidence.

**Figure 7 nanomaterials-09-01101-f007:**
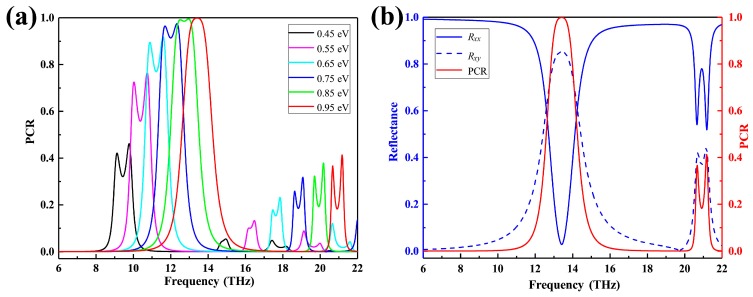
(**a**) The calculated polarization conversion rate (PCR) at different Fermi levels under a normal incidence. (**b**) The magnitude of reflection coefficients *Rxx*, *Rxy* and PCR at the *E_F_* = 0.95 eV.

**Figure 8 nanomaterials-09-01101-f008:**
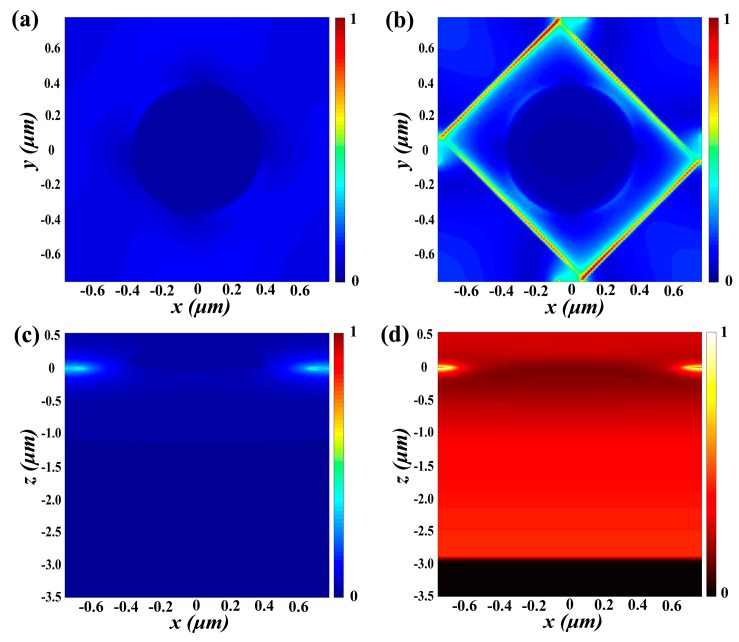
(**a**) |*E*| of the polarization converter at the interface between air and VO_2_ film (**b**) |*E*| of the polarization converter at the interface between VO_2_ and dielectric layer (**c**) |*E*| in the center of the *xz* plane of the polarization converter of the cell (**d**) |*H*| at the *xz* plane of the polarization converter. All frequencies are set at 13.4177 THz.

**Figure 9 nanomaterials-09-01101-f009:**
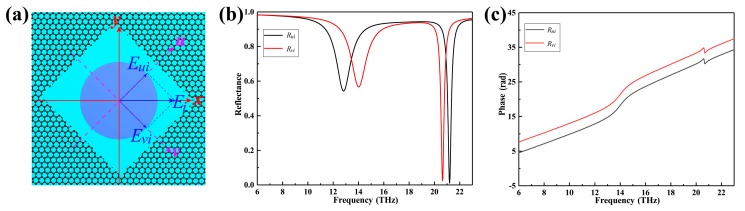
(**a**) Schematic of the decomposition of linearly polarized incident wave. (**b**) The reflectivity of polarized waves along the *v*-axis and *u*-axis. (**c**) The phase of polarized wave along *v*-axis and *u*-axis.

**Figure 10 nanomaterials-09-01101-f010:**
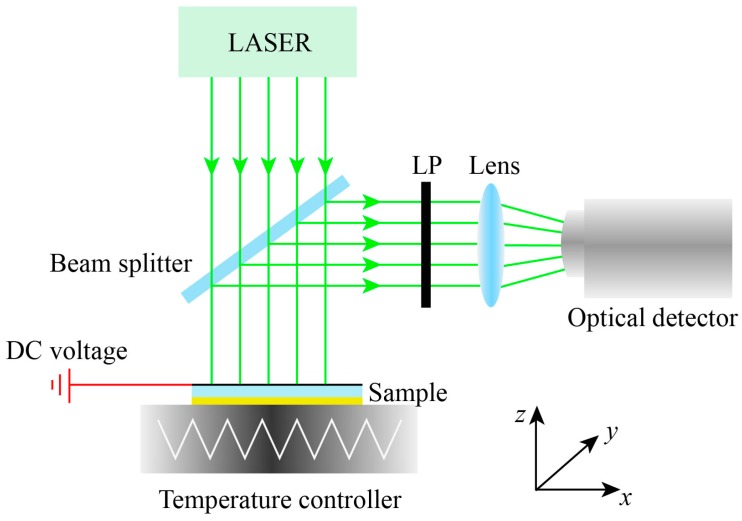
Schematic of the experimental device.

**Table 1 nanomaterials-09-01101-t001:** Comparison of parameters of absorber and polarization converter.

Parameter	Absorber	Polarization Converter
VO_2_ thickness (µm)	0.2	
VO_2_ radius (µm)	0.4	
Size of hole in graphene (µm)	1.0 × 1.2	
Refractive index of dielectric	1.5	
The thickness of the dielectric (µm)	2.9	
Graphene Fermi level (µ = 1 m^2^/Vs)	*E_F_* = 0.1 – 0.95 eV	*E_F_* = 0.95 eV
Permittivity of VO_2_	Drude model	*ε* = 9, σ = 200 S/m

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
