# Peer review of "Dynamically Temperature-Voltage Controlled Multifunctional Device Based on VO_2_ and Graphene Hybrid Metamaterials: Perfect Absorber and Highly Efficient Polarization Converter"

_nanomaterials, 2019, doi:10.3390/nano9081101_

Reviewer 1 Report

The article concerns the design of a new device for terahertz microelectronics. The subject fits the journal profile. Doubts arise from the possibility of obtaining modeled parameters in the real device and the application potential. Detailed doubts are given below:

The Authors claims in the abstract that their results demonstrate that this multifunctional device has widespread applications in emitters, sensors, spatial light modulators, IR camouflages, used in thermophotovoltaics, and wireless communication. Although there is no proof nor discussion in mainbody text.

The Authors must clearly say what is extraordinary in high absorption between 5.7 and 6.8 microns. A lot of materials have high absorption in Mid-IR

The Authors refer Absorption to Reflectance in Figure 2. Have they analysed also scattering?

The authors should demonstrate why they adopted a particular structure geometry for modeling. Did they do any optimization?

The authors only presented the results of simulations for ideal structures? Did they analyze the impact of possible defects and limitations of production technology?

Did the Authors considered any experimental verification of the computer simulations? In most cases real device parameters are significantly lower than those obtained during simulation.

Reviewer 2 Report

Multiple grammar issues throughout. The paper needs a good proof read!

The intro could include more examples of VO2 being used in RF electronics.

The authors have assumed the permittivity and permeability of the VO2 in the terra hertz frequencies which is a downside to this work and the impact of this assumption on calculations should be explained.

Additionally the authors should be aware that for accurate simulation the real and imaginary permeability should not be assumed to be 1 and 0

Detail what FDTD software application (CST, Feko, HFSS, Comsol etc.) was used for the simulations. Details of the meshing setup would also be appreciated.

It makes no sense to be ‘transverse magnetic (TM) polarized and normal to the metamaterial’ surface in Section 3. To be TM and TE polarized is only relevant in Fig. 6.

Figure 2 – Pick one. There is no need to show both given that T = 0.

Fig 3b should include the frequencies used in the absorber (up to 52 THz). A log scale can be used on the y axis for this.

Please explain what you mean by ‘critical coupling condition’ preceding Figure 4.

Figure 5 and 8 – What is the color axis scale? Is it V/m for the |E|? And if so, what is the simulated incidence power? Would it make more sense to normalize this?

Finally, the paper would benefit significantly from measured results.

Author Response

Round  2

Reviewer 1 Report

Subject of the manuscript fits scientific scope of the journal. The article is limited to numerical modelling of the characteristics of the VO2/graphene heterostructure without experimental verification. Noveltyof the paper and its improtance is moderate. The manuscript can be considered for publication.

Author Response

We appreciate sincerely for your warm and efficient efforts that make our work better, here again we have further improved our manuscript regarding the English language style and experimental envision, and hope this correction will meet your approval.

Reviewer 2 Report

Authors have made significant effort to change manuscript. A comment on future planned epxerimental verification measurements should be included.
